# Prevalence of Developmental Dyslexia in Spanish University Students

**DOI:** 10.3390/brainsci8050082

**Published:** 2018-05-08

**Authors:** Carmen López-Escribano, Judith Suro Sánchez, Fernando Leal Carretero

**Affiliations:** 1Department of Research and Psychology in Education, University Complutense of Madrid, 28040 Madrid, Spain; 2Department of Education, University of Guadalajara, 44100 Guadalajara, Mexico; judithsuro@yahoo.com.mx (J.S.S.); fmlcvfug@yahoo.com (F.L.C.)

**Keywords:** dyslexia, reading, spelling, working memory, stress assignment, Spanish, university students

## Abstract

A recent concern in the field of dyslexia studies is the lack of awareness and attention to university students suffering from this condition. If this problem is serious in countries where the relative opacity of the writing system allows for an early detection and, therefore, effective interventions, it is most critical in countries where transparent spelling makes such detection difficult, except in the most severe cases. In Spain, the diagnosis of dyslexia is rare among university-level adults. The present study pursues three aims: (a) to put forward a screening instrument for the detection of university students at risk of dyslexia, (b) to determine the ratio of university students that could be at risk of dyslexia by means of two different procedures, and (c) to create awareness for a disorder that causes hitherto unrecognized difficulties for an important subgroup of the college population. Six hundred and eighty-six university students in four different fields of study within the general area of Social Sciences from a public University in Madrid completed a Spanish-adapted version of a protocol including stress assignment, spelling words and nonwords, and timed phonological working memory of reading and writing task. Results showed that between 1.6% and 6.4% of this population could be at risk of suffering dyslexia. Such risk is not evenly distributed across the four fields of study. As for gender, the first criterion used yields 1.8 males at risk for every female, but the second criterion has as many males as females at risk. Women were significantly better than men in word spelling. Spelling was best predicted by the timed phonological working memory task of reading and writing.

## 1. Introduction

The International Dyslexia Association adopted since 2002 a definition according to which developmental dyslexia, or simply dyslexia, “is a specific learning disability that is neurobiological in origin. It is characterized by difficulties with accurate and/or fluent word recognition and by poor spelling and decoding abilities. These difficulties typically result from a deficit in the phonological component of language that is often unexpected in relation to other cognitive abilities and the provision of effective classroom instruction. Secondary consequences may include problems in reading comprehension and reduced reading experience that can impede growth of vocabulary and background knowledge”, see [1,2]. In the most recent edition of the Diagnostic and Statistical Manual of Mental Disorders, the word ‘dyslexia’ is considered an alternative term for a specific learning disorder with impairment in reading, i.e., in word reading accuracy, reading rate or fluency, and reading comprehension [3] (pp. 66–67). Dyslexia is agreed to be a lifelong condition, even though its particular course and clinical expression vary according to task demands and support provided as well as the range and severity of the individual’s personal abilities and difficulties. Still, the problems that are usually first observed as the child starts learning to read persist, with variation, into adolescence and adulthood [3] (p. 71) or [4].

The manifestations of dyslexia are different in each stage of development. Up to 4 years of age, language development is the variable that best predicts later reading performance. Between 4 and 5 years, phonological awareness is the dominant predictor. Rapid automatized naming, closely associated with reading speed, becomes important as literacy develops [4]. In Spanish children the most typical features of readers with dyslexia are reading speed [5,6,7], and spelling [7,8,9]. Spelling problems are due to the fact that a consistent orthography for reading, as Spanish, may not may not be as consistent for that of writing. This is because different spelling patterns may represent one particular sound. For instance, the “b” sound is represented by the letters “b” and “v”.

According to [3] the main difficulties that dyslexic adults experience are reading speed and correct spelling. Other authors [10,11], added note taking, essay organization and writing difficulties. And, although the vital skill of reading comprehension tends to reach the expected levels in compensating adults [12,13], they seldom attain the same level as ordinary students and in any case perform best when time constraints are not rigid [14]. A French study suggests that the compensation observed in adults is related to substituting morphological strategies for phonological ones [15]. Still, university students with a history of reading problems confess avoiding courses where they have to learn lots of new words [14] and, when confronted with information-gathering in a different language, they cope with the internet more easily than with textbooks [16]. In any case, attending to both decoding and comprehension in this group of students is difficult, a fact that may lead to a poorer performance in some measures [17]. There is also evidence of a deficit in the processing of sentences with a subordinate structure that takes a toll on working memory [18,19]. All in all, in spite of undeniable compensation, there are subtle but real and persistent differences between dyslexic and non-dyslexic adults [20]. Data from studies of English-speaking children with dyslexia who have been followed prospectively support the notion that poor readers in college, the rate of reading as well as facility with spelling may be most useful clinically in differentiating average from poor readers [21]. The results with Spanish-speaking adults are similar: Spanish university students have spelling problems, must constantly checking the orthography, and easily confuse letters when writing [22]. They also experience trouble in phonological awareness and reading speed [23]. 

The phenomenon of dyslexia is common to all languages and writing systems, but it seems to be less frequent the more transparent the orthography is, i.e., the more it has a one-to-one correspondence between graphemes and phonemes [24]. Whilst relative orthographic transparency may hide the condition of dyslexia from view, thus making diagnosis more difficult and less frequent, there is a good chance that prevalence is the same across languages and writing systems. However, the measure of prevalence depends on sampling and definition. According to the Diagnostic and Statistical Manual V (DSM-V) [3], prevalence amounts to 4% among schoolchildren, with a relatively slight, yet significant male predominance from 1.5:1 to 3.1:1 [25]. A common definition sets the cutoff for reading achievement to 1.5 standard deviations below the mean for age and identifies 7% of the population [4].

The prevalence of dyslexia among university student in Sweden was studied by Woff and Lundberg [26]. They used several criteria to measure the prevalence of dyslexia among art students (mean age 27.5) as opposed to non-art students (mean age 24.5), the most restrictive of which classified as dyslexic those students who scored 1 standard deviation below the mean in three tasks: self-report, phonological choice, and word-chains, see [26] (p. 40, Table 2 of the mentioned reference). This criterion indicated that 4% of art students were dyslexic as opposed to only 1.5% among non-art students.

In the last few years, there has been increasing interest about the cognitive profile of university students with dyslexia, especially in non-English-speaking populations [22,23,26,27,28]. There is no doubt that an unknown proportion of university students may suffer from dyslexia. Given that their learning goals are the same as those of their non-dyslexic fellow students, it is vital to be able to identify them, so that they may be informed as to what resources might help them optimize academic outcomes. The academic authorities should also know about useful and justifiable means of support for this class of students, such as extra time during examinations or the use of word processing software with spell checkers [28]. A critical issue in this context is the determination of dyslexia. Deficits in phonological coding continue to characterize dyslexic readers even in adolescence and adulthood [21]. In order to identify university students with dyslexia, we used a variant of the protocol devised for that purpose by Paulesu et al. [24]. This protocol is based on the widely corroborated idea that problems in phonological coding and decoding are the central deficit in dyslexia [29].

In the international study of Paulesu et al. [24], the detection of dyslexia in Italian, French and English university students followed a two-stage procedure. In the first stage, a sample of 1200 students of each nationality were subjected to a collective test of spelling (a list of words and nonwords) and stress assignment (they had to indicate the stressed syllable in a list of polysyllabic words). The 10% with the lowest score, 120 students for each nationality, was selected. In the second stage, the selected subsample was individually given a series of tests in word and nonword reading speed, digit naming, phonological working memory, and spoonerisms. Again, the 10% with the lowest score, 12 students for each nationality, was then considered dyslexic. The 36 students were eventually subjected to a neuroimaging PET study to compare their neurophysiological profile when solving reading tasks. On the basis of that study, the authors concluded that phonological processing is a universal problem underlying reading disorder independently of the degree of transparency of the writing system involved. The purpose of the research was not to find out the prevalence of dyslexia among university students, but rather to establish whether dyslexia is a universal neurobiological condition.

The present study, however, would like to test a screening procedure that might be capable of discriminate university students with dyslexia from the rest of the population. For that purpose, we applied the three tasks used in the first stage of Paulesu et al. [24]: word dictation, nonword dictation, and stress assignment. The first two tasks concern spelling and are motivated by the finding that spelling is one of the major problems facing adults with dyslexia and one of the main characteristics of dyslexia in the Spanish language [8,9,10].

The third task is also important because a study by [30] established that Spanish children with dyslexia had poorer outcomes than the controls in stress assignment tasks. Another recent study confirms the presence of prosodic difficulties in dyslexic students which may depend on subtle auditive deficits [31].

Apart from the three tasks inspired by the first stage of the study of Paulesu et al. [24], we found convenient to add a similar task that the one used by Paulesu et al. [24] in their second stage, that could be administered collectively viz. a timed phonological working memory task of reading and writing. In this task, subjects have to read ten lists of four nonwords, with limited time, store them in working memory, and reproduce them in writing. The rationale for this additional task comes from recent studies which suggest deficiencies relating to working memory, processing speed [32,33], and coding and decoding [29] in students with specific reading disabilities.

As far as we know, there are no studies of working memory and reading in a Spanish-speaking population of university students. Studies done on primary school children [34] and secondary school teenagers [35] show that reading is a highly demanding task that involves the simultaneous processes of extracting and constructing meaning in which working memory’s executive processes play a crucial role. 

The great advantage of the four tasks selected is that they can be collectively administered. Given that they correspond to the main deficits believed to be present in adult populations with dyslexia, they seem to constitute an excellent initial screening procedure that can be used to detect and support Spanish-speaking university students who may be experiencing dyslexia without knowing it.

The three primary aims of this study are:(1)To put forward a protocol to be administered collectively, with sufficient discriminative power, easy to apply to a large number of students, and capable of serving as an initial screening procedure.(2)To establish the proportion of university students in the area of social sciences who might be at risk of dyslexia applying and comparing two different criteria or definition of students at risk of dyslexia.(3)To raise awareness of the educational authorities and the general population of the existence of a hitherto unrecognized problem.

We also have three main questions:

Question 1: How large are the measures of prevalence of risk for dyslexia relatively to the criterion or procedure used?

Question 2: Are the scores obtained by social science students similar or different across fields of study?

Question 3: Which of the tasks included in the study might best explain the persistent spelling mistakes that, according to all observations, most clearly characterize an adult with dyslexia?

## 2. Materials and Methods

### 2.1. Participants

For this study, data of 686 second-year Social Sciences students from a public University of Madrid were used. This group is constituted by four subgroups of second-year students of the School of Law and Business Administration (Law and BA) (140 students; 44 males and 96 females), the School of Journalism (232 students; 64 males and 168 females), the School of Education (177 students; 56 males and 121 females), and the School of Economics (136 students; 68 males and 68 females). The male-female proportion (34–66%) in the present study is quite similar to the proportion reported by Ministerio de Educación, Cultura y Deporte (MECD) (Ministry of Education, Culture and Sport) [36] for university students of Social Sciences studies (40–60%). The total group of 686 students thus consisted of 233 males and 453 females. Mean age was 21.3 years-old with a minimum age of 18.8, and a maximum age of 29.9 years old. Students were told beforehand about the content of the tasks and the amount of time allotted for their solution, and it was explained to them that participation was voluntary, so anybody who did not want to participated could leave the room. The absolute privacy of their personal data was also guaranteed.

### 2.2. Materials

#### 2.2.1. Stress Assignment Task (from Now on SA)

In Paulesu et al.’s study [24], 90 trisyllabic words were chosen, for which the relevant opposition was paroxytonon vs. proparoxytonon (stress on penultimate vs. antepenultimate syllable). This makes sense for Italian because very few Italian words are oxytona (stress on last syllable) and all of them have an obligatory diacritic to mark the fact, e.g., *omertà*, *società*. Again, trisyllabic paroxytona and proparoxytona in Italian very rarely need a lexically differentiating diacritic, e.g., *principi* can mean both “princes” (plural of *principe*) and “principles” (plural of *principio*), so that occasionally we meet the graphic representations *prìncipi* (proparoxyton) vs. *princìpi* (paroxyton), which in any case are optional in Italian. In contrast, the Spanish orthographic rules are utterly different: all proparoxytona must carry a diacritic, whereas both paroxytona and oxytona carry diacritics according to lexical frequency. Thus, most Spanish paroxytona end in a vowel or in one of the two consonants “n” or “s”, whereas most oxytona end in any other of the phonotactically allowed consonants. So, the spelling system prohibits diacritics in such frequent words (e.g., the paroxytona *casa*, *pozo*, *tope*, *comen*, as opposed to the oxytona *azul*, *reloj*). But whenever this dominant stress pattern is broken, i.e., when a paroxytonon ends in a consonant different from “n” or “s” or an oxytonon ends in a vowel or in “n” or “s”, the Spanish spelling has an obligatory diacritic marking the pattern break (e.g., the paroxytona *fácil*, *cárter*, and the oxytona *bebé*, *cebú*). In view of this orthographic difference with Italian, we changed Paulesu’s protocol and set forth a list of 90 printed words which were bisyllabic and had no diacritic. In this way, we reproduced the intention of the original study [24], viz. a binary decision without interference of orthographic rules. The stress pattern of such words cannot indeed be extracted by orthographic information alone and requires phonological recoding of the orthographic string. Subjects were given 3 min to underline the stressed syllable in each word, and the number of correct responses were scored. Split-half reliability is 0.94. See Appendix A.

#### 2.2.2. Spelling Tasks: Word Dictation (from Now on WD) and Nonword Dictation (from Now on NWD)

Here we used the original protocol of Paulesu et al. [24], viz. (a) to write, under dictation, a list of 20 bisyllabic and trisyllabic words, all frequent and familiar, (b) to write, under dictation, 20 nonwords derived from the former list, without infringing the Spanish phonotactic rules. Dictation took place under paced conditions (1 stimulus dictated every 3 s). The stimuli were recorded and reproduced in the same way in each classroom. The number of correct written words was scored. Whenever a word was not written with the orthographically correct diacritic, it was considered an error. Split-half reliability for WD is 0.43 and for NWD 0.48. See Appendix B. 

#### 2.2.3. Timed Phonological Working Term Memory of Reading and Writing Task: Span for Nonwords (from Now on TPWMRW)

In this task subjects have to read ten lists of four nonwords each, to store them in working memory, and to reproduce them in writing. Subjects started their written serial recall at the end of the presentation of each list of four nonwords. The number of correct responses (item identity and item position of the list) were scored. This “timed” task requires students to read, remember, and then write the series of nonwords presented in a video screen. Therefore, the task integrates processing speed, phonological working memory, decoding and encoding, that make the task more complex than a traditional working memory task. In fact, this task captures a complex set of operations that are indeed predictive of dyslexia as we have described in the introduction. Split-half reliability is 0.78. See Appendix C.

All tests were delivered in university classrooms.

## 3. Results

### 3.1. The Prevalence of the Risk of Dyslexia

The primary purpose of this study—associated with our first question—is to determine the prevalence of risk of dyslexia in a group of university students of Social Sciences.

A test, or a battery of diagnostic tasks, like the ones used in this study, aim at sorting the participants into two groups of individuals, one group who exhibits the signs or symptoms underlying the condition or research is about (in this case, developmental dyslexia) and a second group who does not. 

It is not easy to find a feasible screening tool that can be administered to a group to detect dyslexia. Any criterion we may use for the purpose will produce both false positives (non-dyslexic subjects who are pointed out as at risk for dyslexia) and false negatives (subjects at risk for dyslexia who are passed over as non-dyslexic).

Prevalence estimates, of course, depend on definition. A common definition used to identify subjects as having a given condition is to pinpoint those whose score lies 1 or 1.5 Standard Deviation below the mean in a majority of the tasks chosen [4].

In the present study, we propose two definitions or criteria to define subjects at risk of dyslexia: the traditional criterion or definition (those whose scores lies 1 SD below the mean in three of the four tasks) and a second criterion or definition (to find the score distribution in all tasks and then to fix a cut-off point which clearly includes the subjects with the lowest overall scores. This cut-off point is particularly apt for non-normal distributions as it is the case in the present study) and is based on a left-asymmetry of the distribution of the overall scores (See Figure 1 and Figure 2).

First criterion. A student is at risk of dyslexia if he or she scores one standard deviation below the mean for at least three of the four tasks performed in the present study. According to this criterion, 11 out of 686 students (or 1.6% of the sample) qualify as at risk for dyslexia. In absolute numbers we have 6 out of 453 females, 5 out of 233 males, yielding a male-to-female ratio of 1.80:1, which clearly confirms the often observed gender difference in favor of female individuals. As for the distribution by school, 7 students belong to the School of Economics, 3 to the School of Education, 1 to the School of Journalism, and none to the School of Law and BA.

Second criterion. Here we first added the scores of the four tasks for each participant (see Figure 2 for the distributions) and then calculated a cut-off point in the overall distribution (see Figure 1). Note that Figure 1 exhibits a non-normal distribution, where a small group of low-score students is clearly separated from the rest on the left side. The cut-off point is located at 129, about 1.2 SD below the mean. All 44 subjects at or below that score might be at risk for dyslexia (29 females and 15 males), or 6.4% of our sample. Considering that there are 453 females as against only 233 males, the male-to-female ratio is about 1:1. The distribution by school is as follows: 17 belong to Economics, 15 to Education, 8 to Journalism, and 4 to Law and BA.

The second criterion is less restrictive and yields four times as many subjects at risk for dyslexia as the first one, so the question arises whether all individuals picked up by the first criterion are included in the set of those selected by the second one. To answer that question, we calculated the sensitivity (proportion of true positives identified by the second criterion in relation with the first one) and specificity (the proportion of true negatives that are correctly identified by the second criterion in relation to the first one), and mapped the values onto a Receiver Operating Characteristic (ROC) curve (see Figure 3).

In the ROC analysis, the second criterion (sum of scores) would represent the “test” and the first criterion a possible “outcome”. Sensitivity was found to be 0.81 and specificity 0.05, with 0.88 as the Area under the Curve. This means that the second criterion selects 81% of subjects-at-risk and 95% of subjects-without-risk of those selected by the first criterion.

After comparing the results, we suggest using both criteria for identifying students at risk for dyslexia, given that the results of both criteria are consistent with, yet complementary to each other, in such a way that the joint use of both criteria would decrease the number of false positives and false negatives. These procedures might assist to conclude that developmental dyslexia among university students could have a prevalence of between 1.6% and 6.4%.

### 3.2. Descriptive Statistics

Descriptive statistics of the tasks and their comparisons across the different Schools are reported in Table 1.

The comparison of the mean scores obtained in each field of study, we found small differences, with the highest scores in Law and BA and the lowest in Economics. Focusing on prevalence, as measured by our two criteria, we also found that more students in Economics and in Education show risk for dyslexia than those in Journalism or Law and BA. In order to answer our Question 2, we run a One-Way Analysis of Variance (ANOVA). Bonferroni-corrected post hoc analyses, and effect size were also performed in case of multiple comparisons. We found significant differences across all tasks according to field of study, with the lowest scores in Economics and the highest in Law and BA. However, if we consider effect size results, the differences by field of study, whilst significant, are rather small (See Table 2).

### 3.3. Correlation between Reading and Spelling Measures and Sex

Table 3 shows the correlations between the measures in the sample. Inspection of the correlation matrix revealed low but significant positive correlations between WD and all the other tasks. Given the sample size, the correlation is small, but significant. WD was highly correlated with sex, women being significantly better than men in word spelling. NWD correlated with WD, TPWMRW, and sex. TPWMRW did not correlate with sex and SA did not correlate with sex, nor with NWD.

Considering combined spelling efficiency (words and nonwords), significant differences between females and males were found for words (Females M = 19.8; Males M = 19.4) F (1, 685) = 38.25, *p* ≤ 0.01, and for nonwords dictation words (Females M = 12.2; Males M = 11.5) F (1, 685) = 7.17, *p* ≤ 0.01.

### 3.4. Multiple Regression Model

In relation to our Question 3, a multiple regression model was constructed to determine which variable explained variation in WD. NWD, SA, and TPWMRW were chosen as control variables in the model because of their consistent relationship with WD. The results in Table 4 show that TPWMRW explained the most significant portion of the variance in WD.

## 4. Discussion

The main goal of this study was to put forward a screening protocol for the detection of dyslexia among university students. The protocol was designed under the premise that we need it to be simple, easy to administer, and valid. To guarantee validity we adapted a protocol used for a sample of Italian, French, and English students [24], whose measures are supported by previous research as clearly related with dyslexia in university students. Such a user-friendly protocol is badly needed, given the lack of specific tools for detecting dyslexia in this population and given the fact that the dyslexia condition tends to be undetected, even by the sufferers themselves, so that they face difficulties in their studies without receiving any attention or support.

For this study, we chose students from different areas within the social sciences. Although we observed few notable differences across areas, the students of Law and BA had the highest scores, whereas the students of Economics obtained the lowest. Explaining this gradient is not easy, but it may be related to university entry grade requirements.

Entry grade requirements are established in accordance to the relative offer and demand of studying slots in each area. As the number of slots offered goes down and the number of slots demanded goes up, the grade requirement is higher. The highest possible grade is 14. In the academic course of the participants in this study the grade requirements published by the university were 9.23 for Law and BA, 7.22 for Education, 7.03 for Journalism, and 5.89 for Economics [37]. Students with unidentified dyslexia who do the obligatory university entry tests in Spain may be unwittingly discriminated against in that they would be denied entry to those areas with higher grade requirements due to the fact that spelling errors lead to lower scores. They would, thus, be unfairly disadvantaged in choosing a career.

Students who have been identified as dyslexic before doing the entry tests and so are able to present an officially certified diagnosis of dyslexia do enjoy special conditions during university testing. Such special conditions vary according to the laws of the Autonomous Community they belong to. Some of these conditions are: specially adapted tests, bigger fonts, extra time or even no time limit, the use of an electronic calculator or a dictionary, or even in some cases no consideration of spelling errors. Yet, if a dyslexic student has not been officially identified as such, he or she would not be subject to such special conditions and so would be unfairly discriminated against. Hence the importance of having an easy-to-use screening tool whose application to big numbers of students would have a very small cost.

The prevalence of university students at risk for dyslexia which we found in this study, using the first criterion, was 1.6%. If we consider that the total number of university students in the various areas of the Social Sciences in Spain, as reported by the Ministerio de Educación, Cultura y Deporte (MECD) (Ministry of Education, Culture and Sport) [36] in the academic year 2015–2016 was 614, 614, then no less than 9834 students would be at risk for dyslexia and would be suffering from unwitting discrimination if their condition goes undetected. The use of a second criterion to detect dyslexia leads to an estimate of 6.4% prevalence, which is much higher than the 1.6% of the first criterion. The arithmetic difference between the two criteria is explained by the fact that only students having scores lesser than 1 SD in at least three of the four tasks were included according to the first criterion, whereas the second, more general criterion picks up both students who had low scores in several tasks and those who had very low scores in just one of the tasks, so that overall low scores have a different composition in the two cases. It is worth reminding that the tasks have been designed to provide an initial screening only. A later analysis of the two classifications has to be carried out in order to check the specifics of each case.

Observed gender ratios, corrected for differences in the total numbers of males and females in the sample, gives 1.80 males for each female, when using the first criterion (1 SD below the mean in three of the four measures) and a 1:1 ratio after second criterion (adding the scores of the four tasks for each participant and then estimating a cut-off point in the overall distribution). Dyslexia is known to be more common among males than females. However, the magnitude and origin of this gender difference are in doubt. Differences in previously reported ratios, and in the ratios of the present study, may have arisen from differences in the way studies are conducted, namely the adopted definition of dyslexia, the type of test used, the levels of severity of reading troubles required to count as indicating dyslexia, and the sampling error associated with small studies [38].

The scores obtained by women in word and nonword dictation were significantly higher than men’s. Why females generally perform better on language tasks than males is unknown. However, sex differences have been identified even in children as young as 2–3 years old [39]. Although small, female advantages for verbal and written language persist into adulthood [40].

Finally, of all the variables studied, the timed phonological working memory task of reading and writing explains a greater proportion of the variance in spelling (word dictation), which is one of the tasks considered to be best at discriminating for dyslexia in Spanish. As explained in the introduction, recent studies reveal that the cognitive profiles of students with dyslexia are associated with a weak working memory, speed of processing difficulties, coding and decoding (all of the tasks included in the TPWMRW task). We thus conclude that this task should be one of the measures for the proposed initial screening test, together with word and nonword dictation.

It could be objected that the low reliability of the WD and NWD tasks reduce the usefulness of the present study. However, one should not forget that orthographic precision varies according both to each student’s individual difficulties and to the kind of word used. We have tried to counteract this variation by using a very simple task with easy words, which most students can solve it correctly. Therefore, any significant low score obtained in this task probably is an apt indicator of genuine underlying difficulties.

Another potential limitation of this study is that reading fluency, which is highly predictive of dyslexia in Spanish, was not part of our protocol. Still, our goal was to propose a straightforward and easy-to-use screening method that can be applied non-individually to big samples in very little time and at no great expense. We are fully aware that to confirm the presence of dyslexia a second, deeper evaluation should be applied to the individuals hereby identified as possibly at risk, an evaluation that should at least include testing reading precision, reading speed and Intelligence Quotient IQ. For this reason, we have not talked of actually detecting dyslexia, but only the risk for dyslexia. Full, reliable diagnosis will need much more than what we have offered here. However, having a valid and reliable as well as quick and user-friendly means of detecting risk might encourage the educational authorities as well as the staff who teach adult students, at university, college or any other technical level, to be alert to the possibility that some of their charges may be suffering from this condition. At the very least we can fight the widespread lack of awareness as to the nature and consequences of adult dyslexia.

In an effort to improve on the validity of our method, a later study might follow other studies [23,26] by adding a self-report questionnaire, including items on reading-writing difficulties in daily life. The correlation between the score obtained in such a questionnaire and our screening scores might improve the discriminant power of the present procedure. Finally, all students singled out as at risk for dyslexia according to the two criteria should be studied separately so as to gauge the specificity and sensitivity of our task in detecting dyslexia.

## 5. Conclusions

Developmental dyslexia is a lifelong condition, so that people affected by it will continue to suffer from it all their adult lives. In some cases, symptoms might be compensated by certain strategies and a lot of practice, but in other cases such compensatory strategies yield little fruit. Dyslexia is sometimes detected and diagnosed, but not always, so that people affected by it will suffer and do a lot of extra work in order to achieve what is easy for others, yet without knowing why everything is so hard.

The present study proposes a partial remedy to this situation by putting forward a group protocols capable of detecting dyslexia and making an estimate of the number of people affected by this condition. We want thereby to help raise awareness among teachers and the general public about the difficulties experienced by some students. This is the first step in the long-term goal of helping those students themselves.

After doing the study, we have become aware of its limitations and are in a position to suggest measures that might increase the discriminant power of our procedure, whose ultimate purpose is to increase the visibility of developmental dyslexia in adults, so that it can be appropriately conceptualized and treated.

## Figures and Tables

**Figure 1 brainsci-08-00082-f001:**
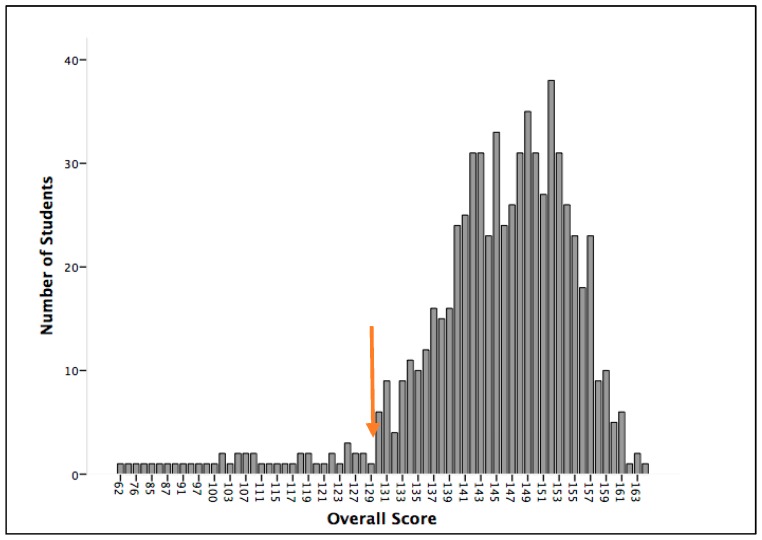
Distribution of Overall Scores (sum of scores in the four tasks). (Mean (M) = 144.3; Standard Deviation (SD) = 12.7). Red arrow indicates selected cut-off score.

**Figure 2 brainsci-08-00082-f002:**
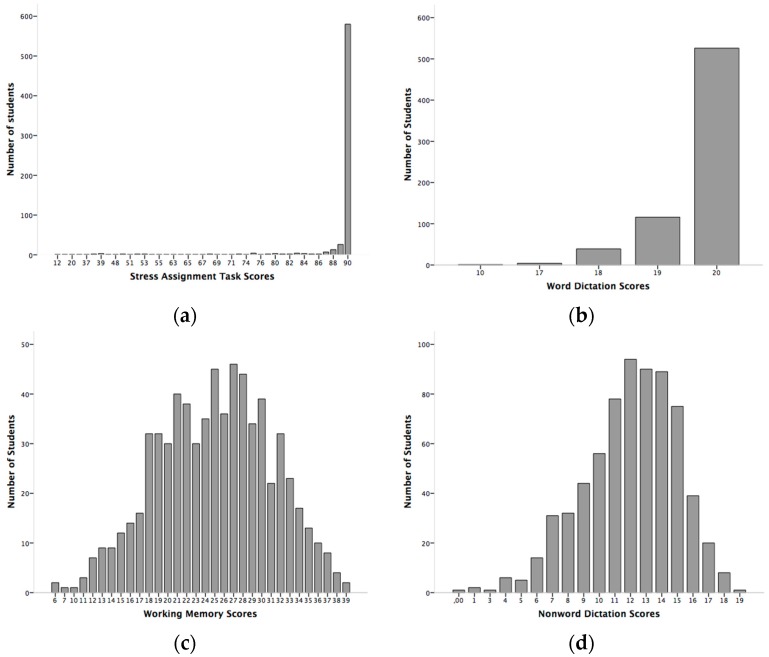
Distributions of Single Task Scores: (**a**) Stress Assignment; (**b**) Word Dictation Scores; (**c**) Working Memory Task; (**d**) Nonword Dictation.

**Figure 3 brainsci-08-00082-f003:**
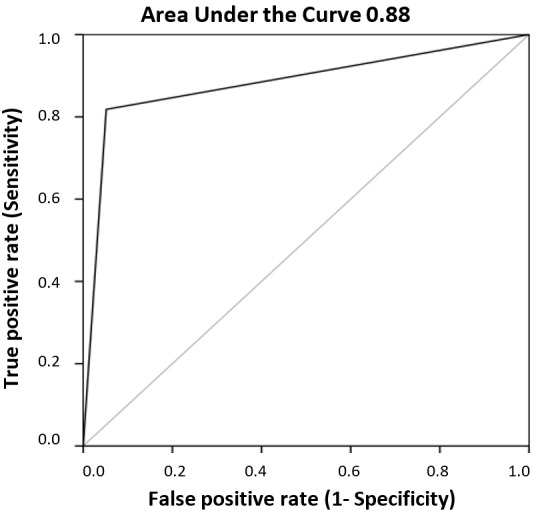
The Receiver Operating Characteristic (ROC) curve resulting from the two criteria used to pick possible subjects at risk for dyslexia.

**Table 1 brainsci-08-00082-t001:** Means and standard deviations of the tasks.

Tasks ^1^	All (*n* = 686)	Lawand BA (*n* = 140)	Journalism (*n* = 232)	Education (*n* = 178)	Economics (*n* = 136)
WD (20)	19.7 (.7) ^2^	19.8 (.4)	19.8 (.5)	19.7 (.6)	19.3 (1.1)
NWD (20)	11.9 (3)	13.7 (1.6)	11.6 (2.8)	11.8 (3.3)	10.9 (3.1)
TPWMRW (40)	24.9 (6.1)	27.3 (4.9)	25.9 (6.3)	23.4 (5.8)	22.8 (5.9)
SA (90)	87.6 (9.2)	88.6 (6.4)	88.4 (7.7)	87.1 (9.5)	85.8 (12.7)

^1^ Maximum score for each task is presented within brackets. ^2^ Standard deviations are presented within brackets. NOTE: WD (Word Dictation); NWD (Nonword Dictation); TPWMRW (Timed Phonological Working Memory of Reading and Writing); SA (Stress Assignment).

**Table 2 brainsci-08-00082-t002:** ANOVA, Post hoc Results and Effect size by Field of Study.

Tasks	F (3, 685)	Post hoc ^1^	Effect Size ^2^
WD	15.9 **	Law and BA = Journalism = Education > Economics	0.06
NWD	26 **	Law and BA > Journalism = Education > Economics	0.10
TPWMRW	19.3 **	Law and BA = Journalism > Education = Economics	0.07
SA	3.1 *	Law and BA = Journalism = Education > Economics	0.01

^1^ Bonferroni. ** *p* ≤ 0.01, * *p* ≤ 0.05. ^2^ Eta-squared. NOTE: WD (Word Dictation); NWD (Nonword Dictation); TPWMRW (Timed Phonological Working Memory of Reading and Writing); SA (Stress Assignment).

**Table 3 brainsci-08-00082-t003:** Correlations between reading and spelling measures and sex (*N* = 686).

Tasks and Sex	1	2	3	4	5
1. WD (20)	-	0.18 **	0.20 **	0.14 **	0.23 **
2. NWD (20)		-	0.17 **	0.07	0.10 **
3. TPWMRW (40)			-	0.13 **	−0.01
4. SA (90)				-	−0.03
5. Sex					-

** *p* ≤ 0.01, * *p* ≤ 0.05. NOTE: WD (Word Dictation); NWD (Nonword Dictation); TPWMRW (Timed Phonological Working Memory of Reading and Writing); SA (Stress Assignment).

**Table 4 brainsci-08-00082-t004:** Multiple regression model.

Step	Variable	R	R^2^ Change	F Change
1	TPWMRW	0.20	0.04	28.4 **
2	NWD	0.25	0.02	28.9 **
3	SA	0.27	0.01	18 **

** *p* ≤ 0.01. NOTE: TPWMRW (Timed Phonological Working Memory of Reading and Writing); NWD (Nonword Dictation); SA (Stress Assignment).

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
