# Peer review of "Prevalence of Developmental Dyslexia in Spanish University Students"

_brainsci, 2018, doi:10.3390/brainsci8050082_

Round 1

Reviewer 1 Report

Thank you for adding the analyses and references. I also think the renaming of the WM variable is helpful. My main concern is that the paper does not address the outcomes of the second criterion, which shows the screening to have low specificity. This is an important finding which should be addressed further. There should also be a section discussing this in the limitations section as well as some suggestions as to how this can be addressed in a "future directions" paragraph. You might suggest, for example, that this screening protocol be tweaked or amended in future studies to improve specificity. What would you suggest changing in a future study? The authors should also address the need for a follow-up validation study in which the screening is tested against Spanish students with confirmed dyslexia diagnoses.

In the discussion, please address further whether these findings are consistent with previous studies (reviewed in the introduction) and the implications of these findings in characterizing dyslexia in Spanish college students in comparison to Italian or French students using similar screening measures (this, I believe, is the main contribution of the study).

I have provided my feedback line-by-line. I have included some English corrections but the paper will need a thorough proofing for grammar before publication.  

Line 27 “Girls were significantly better than boys in word spelling”. Perhaps change to “Women and men” since these are young adults. Please be consistent with these throughout the paper

Line 48 Thank you for providing reliability measures. Split half reliability for the spelling to dictation tasks is quite low. The implications of this finding are important to the goal of the study and should be addressed in the discussion. This is an especially important finding considering that spelling is often used as an indicator or dyslexia in college students, especially in more transparent languages. You've shown here that measured this way, it may not be reliable. This has important implications for your results.

Line 69 “…be so consistent for that of writing” change to “may not be as consistent”

Line 85 In a serial list of infinitives, use “to” only once and then the infinitive form thereafter : to store,  and reproduce….. “to store them in working memory, and to reproduce them in writing.”

Line 99 please provide the language and age for this study: Woff & Lundberg [26].

Line 153 “The present paper takes its departure from a conception of dyslexia as a basic phonological deficit”—this sentence is confusing placed here since you go on to talk about using a screening protocol that is based on phonological measures—consider deleting or moving elsewhere

Line 236-237  "To raise awareness of the educational authorities and the general population of the existence of a hitherto unrecognized problem that concerns a large number of university students." This is an implication of the findings. Stating this here (in the intro) assumes you expect to find a high number of students at-risk for dyslexia, although you haven’t reported these findings yet. It makes the study appear biased. I suggest deleting these and leaving the research questions. Or you could change to say that a goal is simply to prevalence estimates of dyslexia to higher education institutions. 

Line 301 “With their help we might conclude that developmental dyslexia” àPlease reword. Maybe “These tests might assist in ….? “

Line 327 “…of individuals, those with…” Change , to :

Line 330 “In psychology it is not easy to identify a condition by means of test or task batteries related to that condition.” Do you mean that diagnostic tests have no diagnostic utility? I don’t think that’s true. Please revise or delete this sentence. It may be useful to talk here about the difficulty of finding a feasible screening tool that can be administered to a group

Line 395 Thank you for providing the measures. “Sensitivity was found to be .81 and specificity .05” The sensitivity is appropriate for a screening measure, but the low specificity indicates a very high risk that you will over-identify students as having dyslexia who don’t. This should be addressed first at the end of the paragraph (around Line 646) and included as a limitation

Line 645 leds àleads

Line 662 best discriminating à best at discriminating

Line 663 Change “intellectual” profile to “cognitive” profile to be consistent throughout the paper

Line 665 “of them tasks” à”all of the tasks”

All tables should include a Note or caption that defines the acronyms

Author Response

Initial comments:

We have added suggestion for future studies to improve validity of the study.  See lines 124-149.

LINE BY LINE COMMENTS:

ALL REVISED AND DONE

Line 395 Thank you for providing the measures. “Sensitivity was found to be .81 and specificity .05” The sensitivity is appropriate for a screening measure, but the low specificity indicates a very high risk that you will over-identify students as having dyslexia who don’t. This should be addressed first at the end of the paragraph (around Line 646) and included as a limitation

 Sorry, but we are not able to understand this comment.  The area under the curve in our study is .88 that means a good prediction and the specificity must be a low value, as it is the case in the present study.

Thank you for your work and time, and your detailed review that without doubt has improved our paper!

Reviewer 2 Report

The authors have been responsive to the questions and comments I raised in my review of the original submission. 

Author Response

THANK YOU!

Reviewer 3 Report

Review ms "Developmental dyslexia in Spanish university students"

The authors have responded well to the suggestions made.

Author Response

THANK YOU!

Reviewer 4 Report

I'm happy with the paper in its present form

Author Response

THANK YOU!

Reviewer 5 Report

SUMMARY

This paper deals with the prevalence of dyslexia in Spanish adults. The aim is to put forward a screening instrument for the detection of university students with dyslexia. Four different variables were measured (word and no-word spelling, timed phonological working-memory, identification of word stress-pattern) in a sample of 686 university students. Two different criteria were used to assess prevalence, one parametric (SD-difference from the mean) and the other graphical (distributional cut-point). The two criteria give fairly different incidence estimations (1.6 and 6.4 %).

This is an interesting attempt to assess the prevalence of dyslexia in the adult population, which has important theoretical and practical implications. However, I have several concerns and remarks about the methodology.

MAIN CONCERN

The fairly large difference between the prevalence values afforded by the two criteria questions the relevance of each of these criteria. The cause of such difference should be identified and the instrument should be modified accordingly. However, this is not an easy task because there is a twofold difference between the two criteria used to measure prevalence. The first criterion is based on differences in SD relative to the mean (1 SD below the mean) that are measured separately for four different variables. The second criterion is based on a graphical cut-point in the overall distribution of the sum of these four variables. It follows that the difference in the prevalence values afforded by the two criteria might have two different causes.

One possible cause is that the graphical cut-point corresponds to some other difference in SD below the mean in the overall distribution. Taking another SD value (1.5 SD or 2 SD) might then reduce the difference between the two criteria.

Another possible cause is that the variables do not contribute equally to the sum (because of differences in variances). The word-stress scores might contribute more to the prevalence obtained with the SD-based criterion because their variance is larger than those of the two other variables (Table 1), due to distributional left-asymmetries (Figure 2a). The difference between the two criteria might be reduced by using a weighted combination (or Z transforms) of the four variables.

In short, there are several different possible ways to reduce the differences between the “distributional cut point” and “SD-based” prevalence estimations and such possibilities should be systematically explored in order to validate the paradigm proposed in this paper.

However, the “SD-based” prevalence estimation has to cope with the fairly low correlations between the four variables (Table 3: R values not larger than .20, 4% explained variance). This might be a lesser problem for the “distributional cutpoint” estimation if the left-outliers in the distributions of the four variables (Figure 2) corresponded to the same students. This should be investigated.  Finally, still another problem is that no formal procedure is presented to localize the “distributional cutpoint” (see my second comment below).

comments

Lines 96-97 (Introduction): Prevalence of dyslexia among children 4%: Reference to DSM-V [4] should be completed by other sources. (e.g. 7%: Peterson & Pennington, 2012, Lancet).

Lines 339-340 (Results): The definition of second criterion for calculating the incidence is unclear. What does mean “…to fix a cut-off point which clearly includes the subjects with the lowest overall scores.”? What is the criterion used for defining the lowest overall scores?

From the following lines (348-351), it becomes clear that the second criterion is based on a left-asymmetry of the distribution of the overall scores (Figure 1), and it corresponds to the graphical limit between a fairly Normal shaped part and an almost rectangular lower-end distribution. However, no formal procedure is presented to localize this graphical limit.

Lines 645-646 (Discussion): the difference in prevalence estimations between the two criteria is simply mentioned, without further comments. Such difference has paramount implications for the reliability of the instrument and should be followed by an in-depth discussion.

Lines 670-674 (Discussion): the proposal to use the instrument as a tool for detecting the students “at risk” for dyslexia, not those “with” dyslexia is confusing. Detecting dyslexia has important implications for academic achievement and should not be suspended to hypothetical complementary procedures. If the present instrument were accepted for administrative purposes, the subtle distinction between students at risk for dyslexia and those who are indeed affected by dyslexia might be lost in day-to-day practice. With damageable consequences for further developments.

REMARKS

Alternative use of the terms “prevalence” and “incidence”. Should be standardized. Prevalence seems best suited. Incidence refers to a change in prevalence over time.

The object of the paper should be better specified in the title : “Prevalence of Developmental Dyslexia in Spanish University Students”

Line 27: “This variable …” Which one ?: “Sex …”

Lines 105-107: Summarize the outcomes of the [23, 24,26,27, 28] studies cited.

Lines 394-397: There is an error in the account of the ROC analysis  “This means that the second criterion selects 81% of subjects-at-risk and 95% of subjects-without-risk of those selected by the first criterion.”  5%, not 95%.

Figure 1: The authors should give the mean and SD values of the overall distribution

Author Response

MAIN CONCERN:

The arithmetic difference between the two criteria is explained by the fact that only those students having scores lesser than 1 SD in at least three of the four tasks were included according to the first criterion, whereas the second, more general criterion picks up both students who had low scores in several tasks and those who had very low scores in just one of the tasks, so that overall low scores have a different composition in the two cases. It is worth reminding that the tasks have been designed to provide an initial screening only. A later analysis of the two classifications has to be carried out in order to check the specifics of each case.

LINE BY LINE COMMENTS:

Lines 96-97 (Introduction): Prevalence of dyslexia among children 4%: Reference to DSM-V [4] should be completed by other sources. (e.g. 7%: Peterson & Pennington, 2012, Lancet).

DONE

Lines 339-340 (Results): The definition of second criterion for calculating the incidence is unclear. What does mean “…to fix a cut-off point which clearly includes the subjects with the lowest overall scores.”? What is the criterion used for defining the lowest overall scores? 

From the following lines (348-351), it becomes clear that the second criterion is based on a left-asymmetry of the distribution of the overall scores (Figure 1), and it corresponds to the graphical limit between a fairly Normal shaped part and an almost rectangular lower-end distribution. However, no formal procedure is presented to localize this graphical limit. 

DONE.  See lines 502-504.

Lines 645-646 (Discussion): the difference in prevalence estimations between the two criteria is simply mentioned, without further comments. Such difference has paramount implications for the reliability of the instrument and should be followed by an in-depth discussion.

DONE.  See lines 932-939

Lines 670-674 (Discussion): the proposal to use the instrument as a tool for detecting the students “at risk” for dyslexia, not those “with” dyslexia is confusing. Detecting dyslexia has important implications for academic achievement and should not be suspended to hypothetical complementary procedures. If the present instrument were accepted for administrative purposes, the subtle distinction between students at risk for dyslexia and those who are indeed affected by dyslexia might be lost in day-to-day practice. With damageable consequences for further developments.

 The goal of the study is to propose an easy-to-use screening method that can by applied to big samples to detect students that could be at risk of dyslexia.

REMARKS

Alternative use of the terms “prevalence” and “incidence”. Should be standardized. Prevalence seems best suited. Incidence refers to a change in prevalence over time.

DONE. 

The object of the paper should be better specified in the title : “Prevalence of Developmental Dyslexia in Spanish University Students”

DONE. 

Line 27: “This variable …” Which one ?: “Sex …”

DONE.   Spelling

Lines 105-107: Summarize the outcomes of the [23, 24,26,27, 28] studies cited. 

The outcomes of these studies were summarized in the above paragraphs

Lines 394-397: There is an error in the account of the ROC analysis  “This means that the second criterion selects 81% of subjects-at-risk and 95% of subjects-without-risk of those selected by the first criterion.”  5%, not 95%. 

The second criterion includes almost the same subjects-without-risk of those selected by the first criterion, so, the coincidence between the two criteria could not be 5% of the total sample.

Figure 1: The authors should give the mean and SD values of the overall distribution 

 DONE.  

THANK YOU FOR YOUR DETAILED REVIEW AND TIME, WITHOUT DOUBT HAS IMPROVED OUR PAPER